# ENHANCING DEEP TABULAR MODELS WITH GBDT-GUIDED PIECEWISE-LINEAR EMBEDDINGS

## ABSTRACT

Tabular data remains central to many scientific and industrial applications. Recently, deep learning models are emerging as a powerful tool for tabular data prediction, outperforming traditional methods such as Gradient Boosted Decision Trees (GBDTs). Despite this success, the fundamental challenge of feature heterogeneity still remains. Unlike in image or text modalities where features are semantically homogeneous, each tabular feature often carries a distinct semantic meaning and distribution. A common strategy to address the heterogeneity is to project features into a shared high-dimensional vector space. Among the various feature types in tabular data, categorical features are effectively embedded via embedding bags, which assign a learnable vector to each unique category. In contrast, effective embeddings for numerical features remain underexplored. In this paper, we argue that piecewise-linear functions are well suited to modeling the irregular and high-frequency patterns often found in tabular data, provided that breakpoints are carefully chosen. To this end, we propose GBDT-Guided Piecewise-Linear (GGPL) embeddings, a method comprising breakpoints initialization using GBDT split thresholds, stable breakpoint optimization using reparameterization, and stochastic regularization via breakpoints deactivation. Thorough evaluation on 46 datasets shows that applying GGPL to a range of state-of-the-art tabular models consistently improves performance—especially on regression tasks—or at least matches their native numerical embeddings. Together with its negligible overhead, this suggests that GGPL can serve as a practical default numerical embedding for future tabular architectures. The code is available in the supplementary material.

## 1 INTRODUCTION

Tabular data, characterized by its structured format of rows and columns, remains the most common data modality in a vast array of scientific and industrial domains, from healthcare and finance to e-commerce and climate science (Shwartz-Ziv & Armon, 2022; Somvanshi et al., 2024). Its practical importance across numerous domains has established predicting a target column from a set of observed features as a central problem within the field of machine learning, attracting extensive research (Gorishniy et al., 2021; Lee et al., 2024; Eo et al., 2025; Hollmann et al., 2025; Lee et al., 2025; Ye et al., 2025).

Deep learning architectures for tabular data (Yan et al., 2023; Gorishniy et al., 2025; Hollmann et al., 2025; Ye et al., 2025) are increasingly outperforming Gradient Boosted Decision Trees (GBDTs; Chen & Guestrin, 2016; Ke et al., 2017; Prokhorenkova et al., 2018) on many benchmarks, which have long been the dominant approach in the field. This shift is driven by the ability of deep models to capture complex, non-linear feature interactions, reducing the dependence on manual feature engineering. In real-world applications with large and diverse datasets, deep learning models stand out for their high prediction accuracy and efficient inference.

Nevertheless, effectively handling heterogeneous tabular features remains a key challenge for deep models (Gorishniy et al., 2021). Unlike images or text, where features (e.g., pixels or words) are semantically homogeneous, tabular columns often represent fundamentally different concepts whose scales and distributions can vary widely. A common strategy to address this heterogeneity is to map features into a shared high-dimensional vector space, allowing subsequent layers to function

effectively (e.g., matrix multiplication in an MLP or the attention mechanism in a Transformer). Moreover, this projection into a high-dimensional space empowers the network to learn complex, non-linear feature interactions. While there is a well-established practice for categorical features using embedding bags (Mikolov et al., 2013; Guo et al., 2017), a standard methodology for embedding numerical features has yet to emerge.

Embedding a numerical feature can be viewed as learning a continuous mapping from $\mathbb{R}$ to $\mathbb{R}^d$. A variety of approaches have been explored for this purpose, including Multi-Layer Perceptrons (MLPs; Gorishniy et al., 2022; Wu et al., 2024), Fourier features (Gorishniy et al., 2022; Sergazinov et al., 2025), and piecewise-linear functions (Gorishniy et al., 2022). Among these, we find piecewise-linear functions well suited for embedding tabular data. Real-world tabular datasets often exhibit irregular and non-smooth feature–target relationships. A key aspect of these relationships is the presence of high-frequency components, which are critical for accurate prediction (Grinsztajn et al., 2022). However, MLPs exhibit a spectral bias that prevents them from effectively capturing such high-frequency target functions (Rahaman et al., 2019). Although Fourier features have addressed this limitation in computer vision (Tancik et al., 2020; Mildenhall et al., 2021), their effectiveness is reduced in the tabular domain. Accurate modeling with Fourier features requires an appropriate choice of frequency components, but the heterogeneity of tabular data implies that a different set of frequencies may be optimal for each feature. While the expressiveness of piecewise-linear functions in modeling irregular and high-frequency patterns also depends critically on the appropriate placement of breakpoints (Hastie et al., 2009), our method addresses this challenge by effectively selecting suitable breakpoints for each numerical feature.

In this paper, we address the challenge of breakpoint positioning within the prior piecewise-linear embedding (Gorishniy et al., 2022). To this end, we propose the GBDT-Guided Piecewise-Linear (GGPL) embedding that focuses on three essential components: initialization, optimization, and regularization. The properties of tabular data and their target functions make each of these components important. Effectively modeling the irregular and non-smooth function requires precise breakpoint placement, which makes both effective initialization and stable optimization essential. Additionally, the task of tabular prediction exhibits an inherent tendency to overfit (Kadra et al., 2021) and lacks the inherent invariances (e.g., spatial invariance in images) that facilitate data augmentation. This makes robust regularization essential. To address these challenges, we propose the following contributions, which are described in detail in Section 3.

1. We initialize the breakpoints using the split thresholds of the largest gains from an XGBoost model, effectively leveraging the well-established strength of GBDT.

2. We reparameterize the optimization of breakpoints into a stable process by optimizing the ratios of piece lengths on a probability simplex, which guarantees valid breakpoint positions throughout training.

3. We propose a regularization technique that stochastically deactivates breakpoints during training, encouraging similarity between adjacent linear pieces to prevent overfitting.

In practice, we train a default XGBoost once to obtain the split thresholds, which minimizes training overhead and shows no statistically significant difference from a hyperparameter-tuned one. In addition, all proposed methods are training-only and not applied at inference, so there is no additional inference-time overhead compared with prior piecewise-linear embedding methods (Gorishniy et al., 2022).

We validate our proposed method through extensive experiments on the 46 datasets from Gorishniy et al. (2025), spanning a wide range of sizes and domains. Our embedding demonstrates statistically significant improvements in two settings: (i) when applied to state-of-the-art deep tabular models (Yan et al., 2023; Gorishniy et al., 2025; Ye et al., 2025) and (ii) when compared with existing numerical embedding methods (Gorishniy et al., 2022; Li et al., 2024). Notably, our best-performing model achieves the top average rank across all datasets. Moreover, on small-sized datasets where tabular foundation models are available, our GGPL-enhanced models perform competitively, sometimes surpassing TabPFN (Hollmann et al., 2025). The detailed experimental setup and results are presented in Section 4. A subsequent analysis, including ablation studies, statistical tests, and performance analysis across dataset characteristics, is provided in Section 5.

We validate our proposed method through extensive experiments on the 46 datasets from Gorishniy et al. (2025), spanning a wide range of sizes and domains. Our embedding demonstrates statis-

tically significant improvements in two settings: (i) when applied to state-of-the-art deep tabular models (Yan et al., 2023; Gorishniy et al., 2025; Ye et al., 2025) and (ii) when compared with existing numerical embedding methods (Gorishniy et al., 2022; Li et al., 2024). These gains are more consistent on regression tasks, and on classification they are relatively small but at least match the performance of the native numerical embeddings. Combined with the negligible training and inference overhead of GGPL, these results suggest that it can serve as a practical default numerical embedding for future tabular architectures. Notably, our best-performing model achieves the top average rank across all datasets. Moreover, on small-sized datasets where tabular foundation models are available, our GGPL-enhanced models perform competitively, sometimes surpassing TabPFN (Hollmann et al., 2025). The detailed experimental setup and results are presented in Section 4, and subsequent analysis, including ablation studies, statistical tests, and performance analysis across dataset characteristics, is provided in Section 5.

## 2 RELATED WORK

### 2.1 TABULAR PREDICTION MODELS

GBDTs like XGBoost (Chen & Guestrin, 2016), LightGBM (Ke et al., 2017), and Cat-Boost (Prokhorenkova et al., 2018) have long been the state-of-the-art. They build an ensemble of weak decision trees sequentially and are known for their ability to handle sparse, heterogeneous data and capture complex feature interactions. However, their performance is being surpassed by deep learning architectures, which can be largely classified into two main categories: foundation models and task-specific models.

Foundation models like TabPFN (Hollmann et al., 2025) can be applied to various downstream tasks without further parameter tuning, demonstrating remarkable performance on small-scale problems. This is achieved by pre-training a large model on millions of synthetic datasets and leveraging in-context learning at inference-time. However, their quadratic time complexity with respect to the number of training samples restricts their applicability to large-scale datasets.

Task-specific models, while chronologically preceding foundation models, remain a crucial and practical type as they are free from the scalability issues of foundation models. To improve the performance of task-specific models, various approaches have been explored, such as improving backbone architectures, enhancing embedding methods, and incorporating the strengths of GBDTs.

### 2.2 IMPROVEMENTS IN TASK-SPECIFIC MODELS

One primary line of research is designing specialized backbone architectures. These include MLP-based models, which have shown that even simple architectures can achieve top-tier performance when combined with proper regularization and ensemble techniques (Kadra et al., 2021; Holzmüller et al., 2024; Gorishniy et al., 2025); Transformer-based models that adapt the self-attention mechanism to learn complex interactions among heterogeneous features (Gorishniy et al., 2021; Yan et al., 2023); and retrieval-based models that make predictions by retrieving similar instances from the training set (Gorishniy et al., 2024; Ye et al., 2025).

Another key line of research involves improving embedding methods for input features. MLP-based embedding methods apply a feature-specific MLP to map each scalar value to an embedding vector (Guo et al., 2017; Gorishniy et al., 2022; Wu et al., 2024). However, MLPs have a spectral bias towards learning smooth functions (Rahaman et al., 2019), which may not be optimal for the often irregular relationships in tabular data. Inspired by their success in computer vision (Mildenhall et al., 2021), Fourier embeddings have also been used for numerical embeddings (Gorishniy et al., 2022; Sergazinov et al., 2025), but their effectiveness is limited in the tabular domain due to feature heterogeneity, which makes it difficult to find suitable frequency components for all features. Piecewise-linear embedding methods partition a feature's range into a set of bins and learn a linear function within each bin. Gorishniy et al. (2022) introduce piecewise-linear embedding and propose two methods for initializing breakpoints: a quantile-based approach, which places breakpoints based on the input distribution, and a target-aware approach, which trains a decision tree for each input feature to predict the target and uses the resulting thresholds. However, these breakpoints are fixed and not optimized during training. Further, the feature-wise decision tree-based approach prevents considering complex feature interactions when placing breakpoints.

A third line of work incorporates the strengths of GBDTs into deep learning models. Some methods introduce modules to mimic the thresholding behavior of decision trees within a neural network (Popov et al., 2020; Katzir et al., 2021). Another method uses GBDTs to calculate feature frequencies to determine the selection ratio of feature gates (Li et al., 2024). Our approach, rather than mimicking GBDTs, leverages their efficiency and accuracy to initialize the breakpoints of the piecewise-linear embedding using split thresholds with the largest score gain.

# 3 PROPOSED METHOD

## 3.1 BACKGROUND: PIECEWISE-LINEAR ENCODING

To enhance the representational capacity of tabular models, prior work has proposed embedding scalar numerical features into higher-dimensional vector spaces using piecewise-linear encoding (PLE; Gorishniy et al., 2022). Formally, for $i$-th numerical feature $x_i \in \mathbb{R}$, the input range is partitioned into $K_i + 1$ disjoint intervals $[t_k^{(i)}, t_{k+1}^{(i)})$ for $k = 0, \ldots, K_i$, and the encoding is computed as $\mathrm{PLE}(x_i) = [e_0^{(i)}, \ldots, e_{K_i}^{(i)}] \in \mathbb{R}^{K_i+1}$:

$$
e_k^{(i)} = \begin{cases} 0, & x_i < t_k^{(i)} \text{ and } k > 0 \\ 1, & x_i \geq t_{k+1}^{(i)} \text{ and } k < K_i \\ \frac{x_i - t_k^{(i)}}{t_{k+1}^{(i)} - t_k^{(i)}}, & \text{otherwise} \end{cases}
\tag{1}
$$

The conditions on $k$ ($k > 0$ and $k < K_i$) handle linear extrapolation when input values $x_i$ fall outside the range. This encoding produces a continuous and order-aware vector representation of scalar inputs, which can serve as an effective replacement for the original input value in downstream architectures. However, prior implementations of PLE use fixed breakpoints (e.g., quantile or target-aware), which remain static during training. We instead initialize breakpoints using GBDT splits and jointly optimize them via a differentiable reparameterization. This enables the embedding to adapt to high-frequency or irregular patterns. We further apply stochastic regularization to improve generalization.

## 3.2 GBDT-GUIDED PIECEWISE-LINEAR EMBEDDING

The embedding function in a tabular neural network, $\phi_i : \mathbb{R} \to \mathbb{R}^d$, maps $i$-th feature to a shared high-dimensional space. In this paper, we use PLE to model $\phi_i(x_i)$ through a piecewise-linear curve as follows.

$$
\phi_i(x_i) = \left[ \mathbf{w}_0^{(i)}, \mathbf{w}_1^{(i)}, \ldots, \mathbf{w}_{K_i}^{(i)} \right] \begin{bmatrix} e_0^{(i)} \\ e_1^{(i)} \\ \vdots \\ e_{K_i}^{(i)} \end{bmatrix} + \mathbf{b}^{(i)}
\tag{2}
$$

$\phi_i(x_i)$ maps each scalar input to a point on a continuous piecewise-linear curve in $\mathbb{R}^d$. Specifically, when $x_i = t_k^{(i)}$, the embedding becomes a vertex of the curve given by $\mathbf{v}_k^{(i)} = \mathbf{b}^{(i)} + \sum_{j=0}^{k-1} \mathbf{w}_j^{(i)}$. When $x_i \in [t_k^{(i)}, t_{k+1}^{(i)})$, the embedding lies on the line segment connecting $\mathbf{v}_k^{(i)}$ and $\mathbf{v}_{k+1}^{(i)}$. When $x_i < t_0^{(i)}$ or $x_i \geq t_{K_i+1}^{(i)}$, the embedding extrapolates linearly based on the first or last segment, respectively. An example of $\phi_i(x_i)$ is illustrated in Figure 1.

The main challenge is to determine the optimal positions of $t_k^{(i)}$ and their corresponding $\mathbf{v}_k^{(i)}$, which are defined by $\mathbf{w}_k^{(i)}$ and $\mathbf{b}^{(i)}$. The GGPL embedding tackles this through three components:

1. GBDT-guided initialization which determines the initial values of $t_k^{(i)}$,

2. simplex-based reparameterization for stable optimization of $t_k^{(i)}$,

3. stochastic breakpoint regularization to mitigate overfitting.

All parameters including $\mathbf{w}_k^{(i)}$ and $\mathbf{b}^{(i)}$ are optimized via standard backpropagation.

### 3.3 GBDT-Guided Breakpoint Initialization

Our initialization method determines the initial values of $t_k^{(i)}$ using GBDTs trained on the data. Specifically, we adopt the feature threshold values used for splitting nodes in the GBDT as the initial locations for the breakpoints. By leveraging GBDTs' well-established strength, this approach identifies the most effective splits based on their gain scores. This process allocates fewer breakpoints to less important features, reducing the risk of overfitting by saving parameters. Algorithmic details are provided in Appendix A.1.

To keep the pipeline lightweight, we obtain split thresholds from a single XGBoost model with default hyperparameters trained on each dataset. Moreover, because the GBDT is used only to initialize breakpoints, it incurs no inference-time cost. Across the 46 datasets, the tuned XGBoost initializer does not yield statistically significant improvements over the default ($p \approx 0.07$), indicating that the default setting is adequate as a practical choice. Nonetheless, accuracy can still be pushed further through additional tuning of the initializer when desired. Further details are provided in Appendix A.2, including comparisons to tuned XGBoost and alternative GBDT initializers.

### 3.4 Stable Simplex-Based Optimization

Directly training $t_k^{(i)}$ may break ordering constraints ($t_{k-1}^{(i)} \le t_k^{(i)}$) and is prone to division-by-zero errors. To optimize $t_k^{(i)}$ stably, we reparameterize the problem as the following.

We first normalize the position of each $t_k^{(i)}$ as $r_k^{(i)} = (t_k^{(i)} - \min(X_i))/(\max(X_i) - \min(X_i))$, where $X_i$ is the set of values for $i$-th feature in the training set. Then, we define the $k$-th proportion as $\pi_k^{(i)} = r_k^{(i)} - r_{k-1}^{(i)}$. The vector of proportions $\boldsymbol{\pi}^{(i)} = [\pi_1^{(i)}, \ldots, \pi_{K_i+1}^{(i)}]$ forms a point on the $K_i$-dimensional probability simplex ($\Delta^{K_i}$), which is the output of the softmax function.

$$\boldsymbol{\pi}^{(i)} = \text{softmax}(\mathbf{z}^{(i)}) \tag{3}$$

Optimizing unconstrained logits $\mathbf{z}^{(i)} \in \mathbb{R}^{K_i+1}$ instead of $t_k^{(i)}$ guarantees that the breakpoints remain ordered and prevents them from collapsing within the feature's range.

### 3.5 Stochastic Breakpoint Regularization

Since there are no smoothness constraints between adjacent embedding vectors, the learned function exhibits high tortuosity, which increases the risk of overfitting. To mitigate this, we introduce a regularization technique analogous to dropout that encourages similarity between adjacent embedding vectors. During each training forward pass, we randomly deactivate a fraction of $t_k^{(i)}$ for $k = 1, \ldots, K_i$ with probability $p$.

Figure 2 provides a visual example of the stochastic regularization technique. When $t_k^{(i)}$ is deactivated, its corresponding $\mathbf{v}_k^{(i)}$ is ignored, and a new linear piece is formed between $\mathbf{v}_{k-1}^{(i)}$ and $\mathbf{v}_{k+1}^{(i)}$ (or the nearest active breakpoints if multiple consecutive breakpoints are deactivated). The embedding should remain consistent even if some breakpoints are deactivated. This encourages the model

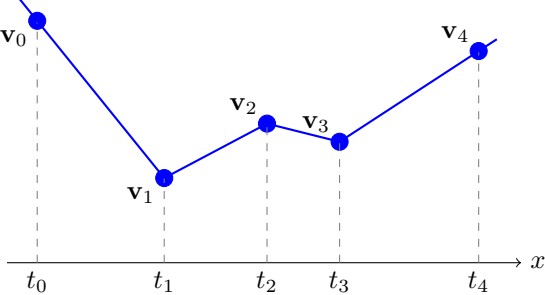

Figure 1: A piecewise-linear embedding that maps a scalar feature into a high-dimensional space, defined by a set of breakpoints ($t_k \in \mathbb{R}$) and their corresponding embedding vectors ($\mathbf{v}_k \in \mathbb{R}^d$).

to learn a smoother function, which is beneficial for regression but can be detrimental for classification tasks that rely on sharp decision boundaries. Consequently, we apply this regularization only to regression tasks. At inference-time, all breakpoints are activated ($p = 0$), and unlike dropout, no scaling of the embeddings is required. We analyze the effect of this regularization for regression and classification in Appendix A.3.

## 4 EXPERIMENTS

### 4.1 DATASETS

We conduct a comprehensive evaluation on the benchmark of 46 datasets previously used in Gorishniy et al. (2025). These datasets span a wide range of tabular tasks, with sample sizes from a few thousand to over a million and feature counts up to nearly a thousand—reflecting the scale and complexity of real-world applications. The characteristics of these datasets are summarized in Table 1, with further details available in Appendix B.

Table 1: Overview of the 46 benchmark datasets, categorized by task, sample size, and feature-to-sample ratio.

| Category | Criteria | Count |
|---|---|---|
| Total | | 46 |
| Task | Classification | 18 |
| | Regression | 28 |
| Sample size | Small ($\leq 30k$) | 24 |
| | Large ($> 30k$) | 22 |
| Feature-to-sample ratio | High ($> 0.001$) | 21 |
| | Low ($\leq 0.001$) | 25 |

### 4.2 BASELINE MODELS

To evaluate the effectiveness and versatility of our GGPL embedding, we integrate it into three state-of-the-art deep tabular models and a baseline MLP, with each representing a different architectural paradigm. For each model, we then compare the performance of the original version against its GGPL-enhanced counterpart. The selected models are MLP, T2G-Former, ModernNCA, and TabM, and we provide detailed descriptions of these four models in Appendix C.1.1.

In addition, we include 10 additional models, including GBDTs (Chen & Guestrin, 2016; Ke et al., 2017; Klambauer et al., 2017; Prokhorenkova et al., 2018; Gorishniy et al., 2021; Somepalli et al., 2021; Wang et al., 2021; Chen et al., 2023; 2024; Gorishniy et al., 2024) for comparison purposes without incorporating our proposed method. TabPFN (Hollmann et al., 2025) is also included in

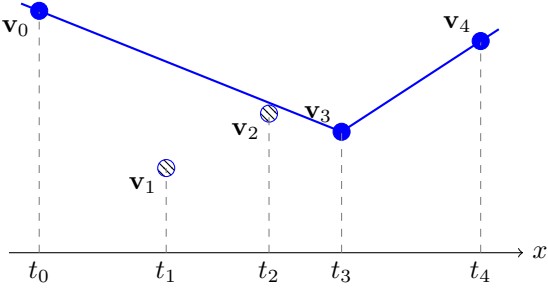

Figure 2: The effect of stochastic breakpoint regularization. When the middle breakpoints are deactivated (dashed circle), a new, linear piece is formed directly between its neighbors (solid circle).

Table 2: Average ranks across 46 datasets (lower is better). The numbers in parentheses indicate the rank improvement from applying GGPL.

| Model | All Tasks (↓) | Regression (↓) | Classification (↓) | Num. Embedding |
|---|---|---|---|---|
| ***GBDT models*** | | | | |
| LightGBM | 9.48 | 8.86 | 10.44 | - |
| XGBoost | 9.00 | 8.93 | 9.11 | - |
| CatBoost | 8.04 | 7.64 | 8.67 | - |
| ***Deep learning models (without numerical embedding)*** | | | | |
| DCN2 | 16.04 | 15.93 | 16.22 | - |
| SNN | 15.20 | 15.61 | 14.56 | - |
| ***Deep learning models (with numerical embedding)*** | | | | |
| ExcelFormer | 13.61 | 13.86 | 13.22 | GLU |
| SAINT | 12.46 | 13.07 | 11.50 | MLP |
| FT-Transformer | 11.70 | 12.25 | 10.83 | Linear |
| Trompt | 10.63 | 10.39 | 11.00 | Linear |
| MLP | 10.28 | 10.32 | 10.22 | Periodic |
| T2G-Former | 9.02 | 9.14 | 8.83 | Linear |
| TabR | 7.98 | 8.32 | 7.44 | Periodic |
| ModernNCA | 7.70 | 8.82 | 5.94 | Periodic |
| TabM-mini | 3.61 | 3.00 | 4.56 | Piecewise-linear |
| ***Deep learning models (with GGPL embedding)*** | | | | |
| MLP-GGPL | 8.28 (-2.00) | 7.64 (-2.68) | 9.28 (-0.94) | GGPL (Ours) |
| T2G-Former-GGPL | 7.80 (-1.22) | 7.07 (-2.07) | 8.94 (+0.11) | GGPL (Ours) |
| ModernNCA-GGPL | 7.09 (-0.61) | 8.07 (-0.75) | 5.56 (-0.38) | GGPL (Ours) |
| TabM-mini-GGPL | **2.96** (-0.65) | **2.04** (-0.96) | **4.39** (-0.17) | GGPL (Ours) |

the comparison only on small-scale datasets that meet its constraints. Detailed descriptions of these baseline models are in Appendix C.1.2.

## 4.3 IMPLEMENTATION DETAILS

For TabM, we use the official implementation from Gorishniy et al. (2025), while our implementations of T2G-Former and ModernNCA are based on the code from Liu et al. (2024). To ensure a fair comparison, we follow the training protocol of Gorishniy et al. (2025).

With some exceptions, we apply a slightly modified version of the quantile transform from scikit-learn (Pedregosa et al., 2011) to numerical features. We use cross-entropy loss for classification and mean squared error loss for regression. Hyperparameters are tuned using Optuna (Akiba et al., 2019) over 100 trials (50 for large datasets). All baselines except TabPFN are also tuned using the hyperparameter search spaces from Gorishniy et al. (2025), while TabPFN is evaluated with default configuration. Further details are provided in Appendix C.2.

## 4.4 RESULTS

For each dataset, we evaluate model performance using accuracy for classification and root mean squared error (RMSE) for regression, averaging the results over 15 random seeds. To aggregate performance across datasets, we compute the average rank of each model based on these scores. Detailed results for each dataset can be found in Appendix D.

**Main Results**

We present the main results in Table 2, which shows the average ranks of our GGPL-enhanced models against various baselines across all 46 datasets, as well as separate ranks for regression and classification tasks. The results consistently show that applying our GGPL embedding leads to

Table 3: Comparison with TabPFN on 23 small-scale datasets.

| Model | All ($\downarrow$) | Reg. ($\downarrow$) | Cls. ($\downarrow$) |
|---|---|---|---|
| LightGBM | 10.91 | 10.40 | 11.88 |
| XGBoost | 10.70 | 10.80 | 10.50 |
| MLP | 10.96 | 11.60 | 9.75 |
| T2G-Former | 10.22 | 9.87 | 10.88 |
| ModernNCA | 9.39 | 10.73 | 6.88 |
| MLP-GGPL | 9.04 | 8.53 | 10.00 |
| T2G-Former-GGPL | 8.57 | 7.20 | 11.13 |
| CatBoost | 8.00 | 7.93 | 8.13 |
| ModernNCA-GGPL | 7.52 | 8.13 | 6.38 |
| TabPFN | 5.96 | 6.93 | **4.13** |
| TabM-mini | 4.57 | 3.47 | 6.63 |
| TabM-mini-GGPL | **3.30** | **2.47** | 4.88 |

Table 4: Ablation study on the components of GGPL. Performance is measured by the average rank on all 46 datasets using the MLP backbone.

| Embedding Method | Average Rank |
|---|---|
| Base | 9.76 |
| Base+I | 8.91 (-0.85) |
| Base+I+O | 8.52 (-0.39) |
| Base+I+O+R (GGPL) | 8.28 (-0.24) |

performance improvements across all four backbone architectures with diverse design paradigms. For instance, GGPL provides a substantial boost to MLP on regression tasks, improving its average rank by 2.68. Overall, TabM-mini-GGPL emerges as the best-performing model, achieving the top average rank of 2.96 across all tasks. This demonstrates that our proposed embedding is not only effective but also versatile, enhancing the capabilities of diverse model architectures.

**Comparison on Small-Scale Datasets**

To motivate our focus on task-specific models, we compare GGPL-enhanced variants against the foundation model TabPFN (Hollmann et al., 2025). This comparison uses 23 small-scale datasets from the full benchmark of 46 datasets that satisfy TabPFN's constraints ($\leq 10$ classes, $\leq 500$ features, and $\leq 10000$ samples). As shown in Table 3, our GGPL-enhanced models are competitive with both TabPFN and CatBoost, even on small-scale datasets where these models are presumed to excel. Notably, TabM-mini-GGPL surpasses TabPFN to achieve the top overall rank. These results highlight that task-specific models can attain state-of-the-art performance without the scalability constraints of foundation models.

## 5 ANALYSIS

### 5.1 ABLATION STUDY

To isolate the contribution of each of our proposed components—Initialization (I), Optimization (O), and Regularization (R)—we conduct an ablation study on the MLP backbone. Starting from a piecewise-linear embedding where breakpoints are initialized uniformly (Base), we incrementally add I, O, and R. The results in Table 4 demonstrate that each component consistently improves performance, validating our design choices.

### 5.2 STATISTICAL TESTING OF GGPL AGAINST OTHER EMBEDDINGS

To evaluate the effectiveness of our proposed GGPL embedding, we conduct two comparisons: (i) against existing numerical embeddings on the MLP backbone and (ii) as a drop-in replacement within other architectures (e.g., T2G, MNCA, TabM), replacing their native numerical embeddings. The baseline embeddings on the MLP backbone include no embedding, periodic encoding, and piecewise-linear embeddings with both initialization methods (quantile-based and target-aware; Gorishniy et al., 2022); additionally, we compare against the tree-based T2V method (Li et al., 2024) on the 16 binary classification tasks.

We apply a stratified Wilcoxon signed-rank test to assess statistical significance across datasets with multiple random seeds: each dataset is treated as a stratum, seed-level paired differences are computed within each dataset, and the stratum-specific statistics are aggregated into a single $p$-value. As shown in Tables 5 and 6, GGPL demonstrates consistent and statistically significant improvements—over all numerical embedding baselines on MLP and over the native numerical embeddings on other architectures.

Table 5: GGPL vs other numerical embeddings on the MLP backbone using stratified Wilcoxon signed-rank test.

| Baseline Method | Z-statistics | $p$-value |
|---|---|---|
| No Embedding | 17.0 | $< 10^{-10}$ |
| T2V | 10.6 | $< 10^{-10}$ |
| Periodic | 3.38 | $7.15 \times 10^{-4}$ |
| Piecewise-linear (quantile-based) | 4.22 | $2.43 \times 10^{-5}$ |
| Piecewise-linear (target-aware) | 4.32 | $1.58 \times 10^{-5}$ |

Table 6: GGPL vs native numerical embeddings across models using stratified Wilcoxon signed-rank test.

| Model | Z-statistics | $p$-value |
|---|---|---|
| MLP | 3.38 | $7.15 \times 10^{-4}$ |
| T2G | 4.90 | $9.74 \times 10^{-7}$ |
| MNCA | 2.37 | $1.79 \times 10^{-2}$ |
| TabM | 4.47 | $7.66 \times 10^{-6}$ |

We further quantify the effect size of GGPL when used as a drop-in replacement for native numerical embeddings by computing Elo ratings (Elo, 1967), which is recently adopted in TabArena (Erickson et al., 2025). Following TabArena, we estimate 95% confidence intervals via 200-round bootstrap resampling (2.5–97.5% quantiles). Unlike TabArena, we calibrate the Elo scale by fixing the MLP backbone with its native numerical embedding to 1000 and report all other Elo scores relative to this anchor.

As shown in Figure 3, GGPL increases the Elo rating of every backbone. For MLP, the 95% confidence intervals of the native and GGPL variants do not overlap. For T2G-Former and TabM-mini, the intervals partially overlap, and the mean Elo of the GGPL variant lies near the upper end of the native variant's confidence interval. For ModernNCA, the intervals overlap more substantially, indicating a smaller yet positive shift. This pattern is consistent with our stratified Wilcoxon signed-rank tests, where ModernNCA also exhibited the largest (yet still significant) $p$-value of $1.79 \times 10^{-2}$. Taken together, the Elo and Wilcoxon analyses confirm that replacing native numerical embeddings with GGPL yields consistent and statistically meaningful improvements across architectures. Further details of the Elo rating and plots of all models are provided in Appendix E.1.

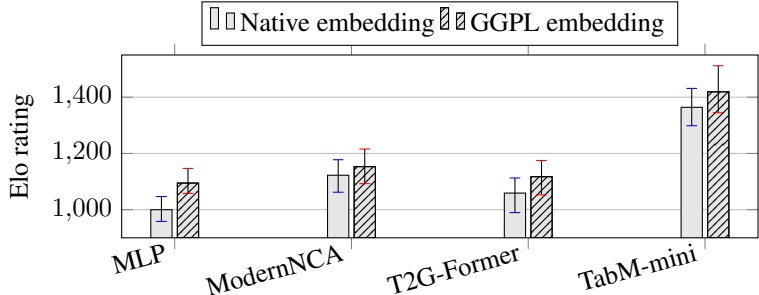

Figure 3: Elo ratings for each backbone with its native numerical embedding (solid bars) and with GGPL (hatched bars). MLP with native embedding is used as baseline (1000). Error bars indicate 95% confidence intervals.

## 5.3 PERFORMANCE ANALYSIS BY DATASET CHARACTERISTICS

To better understand where GGPL provides the most significant benefits, we analyze its performance improvement across different dataset characteristics in Table 7. A few general trends emerge from the data. First, we observe that the performance gains are consistently more pronounced in regression tasks than in classification. In contrast to classification tasks, where precise modeling near decision boundaries is important, regression tasks require global accuracy. This result suggests that our approach is particularly effective for regression tasks, as it helps model the entire feature–target relationship with high fidelity. Second, with the exception of MLP on feature-to-sample ratio, our method tends to yield greater improvements on datasets with small sample sizes and high feature-to-sample ratios. This indicates that GGPL provides a valuable inductive bias that is effective in preventing overfitting, where training data is limited or feature dimensionality is high.

Table 7: Average rank improvement of GGPL across different dataset characteristics.

| Characteristic | MLP | T2G-Former | ModernNCA | TabM-mini |
|---|---|---|---|---|
| Regression task | $10.32 \rightarrow 7.64$ (**-2.68**) | $9.14 \rightarrow 7.07$ (**-2.07**) | $8.82 \rightarrow 8.07$ (**-0.75**) | $3.00 \rightarrow 2.04$ (**-0.96**) |
| Classification task | $10.22 \rightarrow 9.28$ (-0.94) | $8.83 \rightarrow 8.94$ (+0.11) | $5.94 \rightarrow 5.56$ (-0.38) | $4.56 \rightarrow 4.39$ (-0.17) |
| Small sample size | $10.58 \rightarrow 8.21$ (**-2.37**) | $9.38 \rightarrow 8.04$ (**-1.34**) | $8.25 \rightarrow 6.75$ (**-1.50**) | $3.75 \rightarrow 2.92$ (**-0.83**) |
| Large sample size | $9.95 \rightarrow 8.36$ (-1.59) | $8.64 \rightarrow 7.55$ (-1.09) | $7.09 \rightarrow 7.45$ (+0.36) | $3.45 \rightarrow 3.00$ (-0.45) |
| High feature-to-sample ratio | $10.19 \rightarrow 8.52$ (-1.67) | $9.05 \rightarrow 7.67$ (**-1.38**) | $9.10 \rightarrow 7.62$ (**-1.48**) | $3.90 \rightarrow 3.24$ (**-0.66**) |
| Low feature-to-sample ratio | $10.36 \rightarrow 8.08$ (**-2.28**) | $9.00 \rightarrow 7.92$ (-1.08) | $6.52 \rightarrow 6.64$ (+0.12) | $3.36 \rightarrow 2.72$ (-0.64) |

## 5.4 ANALYSIS OF STOCHASTIC BREAKPOINT REGULARIZATION

We investigate the effect of stochastic breakpoint regularization by varying its deactivation ratio ($p$) using the MLP-GGPL model on the House 16H dataset (Gorishniy et al., 2024). While holding all other hyperparameters constant, we vary $p$ from $0.0$ to $0.95$ in increments of $0.05$, averaging results over $100$ random seeds. We evaluate model performance and embedding complexity, where the latter is measured by tortuosity. Further details about tortuosity are provided in Appendix E.2.

Figure 4a shows that RMSE is minimized at $p = 0.60$, indicating that an appropriate level of regularization is essential for the best performance. Figure 4b shows that tortuosity decreases monotonically with $p$, confirming that the regularization smooths the embedding function as intended.

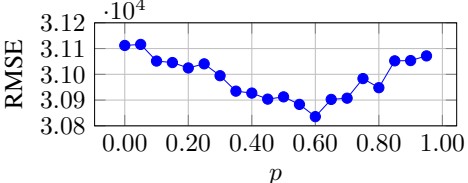

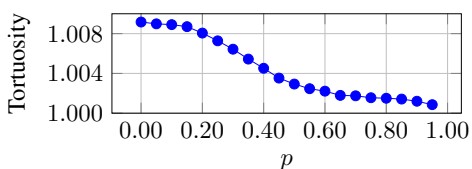

(a) RMSE: Model performance (RMSE) as a function of the deactivation ratio ($p$).

(b) Tortuosity: Embedding function tortuosity as a function of the deactivation ratio ($p$).

Figure 4: The effect of the deactivation ratio ($p$) in stochastic breakpoint regularization.

## 5.5 EFFECT SIZE ANALYSIS

Finally, we aggregate effect sizes across datasets by normalizing each dataset so that the best-performing model scores 1 and the worst scores 0. We then use these normalized scores to compute, for each architecture and task, the mean and standard deviation of the gap between the baseline and its GGPL-enhanced variant. In addition, we plot histograms of the baseline and GGPL-normalized scores to visualize the distributional shifts; detailed statistics and figures are provided in Appendix E.3. On regression tasks, the GGPL distributions tend to shift toward 1 and the mean gains are consistently positive across architectures, whereas on classification tasks the two histograms almost overlap. These results indicate that GGPL improves (on regression) or at least matches (on classification) the native numerical embeddings of several state-of-the-art tabular architectures, supporting our goal of providing a practical default numerical embedding that can be plugged into diverse models with minimal overhead.

## 6 CONCLUSION

In this paper, we addressed the challenge of numerical feature embedding for deep tabular models. To this end, we proposed GGPL, a piecewise-linear embedding method built on three components: GBDT-guided initialization, stable optimization on a probability simplex, and stochastic breakpoint regularization. Our analysis confirms that all three components are essential for the method's effectiveness. With their synergy, GGPL significantly boosts various state-of-the-art models to achieve the top average rank in extensive experiments. Our method demonstrates particular strength in data-scarce settings, indicating it provides a valuable inductive bias to promote better generalization for deep tabular models.

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

## DISCLOSURE OF LLM ASSISTANCE

This paper benefited from assistance by a large language model (LLM) to improve its grammar.

## A ALGORITHMIC DETAILS

### A.1 GBDT-GUIDED BREAKPOINT INITIALIZATION

Given a trained GBDT, we collect node split thresholds and aggregate their gains to find a small set of informative breakpoints for each numerical feature, as described in Algorithm 1.

### A.2 SENSITIVITY TO THE CHOICE OF GBDT INITIALIZER

In practice, we initialize breakpoints by training a default XGBoost model once per dataset. This choice keeps the pipeline simple with negligible computational overhead while already yielding competitive performance. To examine the sensitivity to the choice of GBDT initializer, we compare this default XGBoost against several alternatives on the MLP-GGPL backbone: a tuned XGBoost obtained via a 100-trial hyperparameter search using Optuna (Akiba et al., 2019), a default Light-GBM model, and XGBoost variants with fewer trees ($n_{\text{estimators}}$). All other training settings are kept identical except for the GBDT initializer.

As summarized in Table 8, the tuned XGBoost tends to perform better than the default, but the difference is not statistically significant at the conventional $\alpha = 0.05$ level. Likewise, default Light-GBM and a moderately shallower XGBoost yield performance that is statistically indistinguishable from the default XGBoost initializer, whereas only an extremely small XGBoost (with very few trees) shows a clear degradation. Overall, these results suggest that our default XGBoost initializer is a reasonable trade-off between simplicity and performance, while more aggressive tuning or alternative GBDTs can be used when additional performance gains are desired.

---

**Algorithm 1** Breakpoint Selection from GBDT

---

**Input:** $S = [(i, t, g)]$: A list of node split information from the trained GBDT model, representing feature index, threshold, and gain for each node.
   $X$: The training data, to extract min/max values.
   $K$: Total number of internal breakpoints.
   $N_{num}$: Set of numerical feature indices.
**Output:** $T$: A dictionary mapping numerical features to their sorted list of breakpoints.
 1: $G \leftarrow$ defaultdict(float)
 2: $T \leftarrow$ defaultdict(list)
 3: **for** $(i, t, g) \in S$:
 4:     $G[(i, t)] \mathrel{+}= g$                                    ▷ *Aggregate gain of thresholds*
 5: $G \leftarrow \text{top\_k}(G, K)$                                    ▷ *Select top K splits*
 6: **for** $(i, t) \in \text{keys}(G)$:
 7:     $T[i]$.append($t$)                                    ▷ *Map thresholds to features*
 8: **for** $i \in N_{num}$:
 9:     $T[i]$.extend($[\min(X[i]), \max(X[i])]$)
10:     $T[i]$.sort()                                    ▷ *Add boundaries and sort thresholds*
11: **return** $T$

---

Table 8: Comparison of the default XGBoost initializer with alternative GBDT initializers.

| Comparison | Z-statistic | $p$-value |
|---|---|---|
| vs XGBoost (tuned, 100 trials) | $-1.82$ | 0.067 |
| vs LightGBM (default) | 0.88 | 0.375 |
| vs XGBoost ($n_{\text{estimators}} = 10$) | 0.56 | 0.571 |
| vs XGBoost ($n_{\text{estimators}} = 1$) | 4.14 | $3.34 \times 10^{-5}$ |

### A.3 EFFECT OF STOCHASTIC BREAKPOINT REGULARIZATION

We quantify the effect of stochastic breakpoint regularization by comparing MLP-GGPL models trained with and without the regularizer, separately for regression and classification tasks. For each dataset, we use identical hyperparameters except for the regularization switch ($p > 0$ vs. $p = 0$), evaluate 15 random seeds, and apply a stratified Wilcoxon signed-rank test over seed-level paired differences.

Table 9: Stratified Wilcoxon signed-rank tests for stochastic breakpoint regularization (on vs. off) on MLP-GGPL across 46 datasets.

| Task | $Z$-statistic | $p$-value |
|---|---|---|
| Regression | 1.36 | 0.17 |
| Classification | $-0.17$ | 0.87 |

For regression, the positive $Z$-statistic with a moderate $p$-value suggests a weak trend that the regularized variant tends to perform better than its unregularized counterpart. For classification, the test statistic is slightly negative with a large $p$-value ($p \approx 0.87$), indicating that the regularization does not help for classification. Consistent with these observations, we enable stochastic breakpoint regularization only for regression tasks and disable it for classification.

## B DATASET DETAILS

### B.1 PREPROCESSING

To ensure a fair comparison and reproducibility, we follow the preprocessing used in Gorishniy et al. (2025), from which we adopt the benchmark datasets. Our preprocessing follows their methodology without any modifications. The key procedures are summarized below.

- Numerical features: By default, a slightly modified version of the quantile transform from scikit-learn (Pedregosa et al., 2011) is applied, which adds a small Gaussian noise (mean: 0, std: 1e-5) before calculating the distribution. For exceptions where quantile transform is detrimental, standard normalization or identical mapping is used.

- Categorical features: All categorical features are processed using one-hot encoding.

- Binary features: Features with only two distinct values are mapped to $\{0, 1\}$.

Please refer to the config files on the source code in the supplementary materials for dataset-specific details.

### B.2 DATASET CHARACTERISTICS

We provide a detailed overview of the 46 datasets used in our evaluation in Table 10. This benchmark, originally used by Gorishniy et al. (2025), is composed of datasets from three sources: 28 from Grinsztajn et al. (2022), 10 from Gorishniy et al. (2024), and 8 from Rubachev et al. (2025). The table summarizes key characteristics for each dataset, including its size, feature composition (numerical, binary, and categorical), task type, and its corresponding reference within the benchmark.

## C EXPERIMENTAL DETAILS

### C.1 BASELINE MODEL DETAILS

#### C.1.1 BACKBONE MODELS FOR GGPL

We integrated our proposed GGPL embedding into four backbone models. For each model, the original component for processing numerical features was replaced by GGPL.

- **MLP:** A standard multi-layer perceptron, often used as a deep learning baseline due to its small and lightweight architecture. Gorishniy et al. (2022) compare various numerical embeddings on MLPs and find that piecewise-linear and periodic embeddings yield substantial performance improvements. We use a standard MLP as a primary backbone for evaluating GGPL and the base model for our in-depth analyses.

- **T2G-Former:** T2G-Former (Yan et al., 2023) is a Transformer-based architecture for tabular data that uses a T2G module to model feature interactions. We replace its original linear embedding layer for numerical features with GGPL.

- **ModernNCA:** ModernNCA (Ye et al., 2025) is a retrieval-augmented model that learns a distance metric for nearest-neighbor-based prediction. Its original numerical embedding, based on periodic functions, is replaced with GGPL.

- **TabM:** TabM (Gorishniy et al., 2025) is an MLP-based model with parameter-efficient ensembling. Among its variants, TabM-mini with a piecewise-linear embedding achieves the best performance. In our experiments, we employ the TabM-mini and replace the original quantile-based embedding with fixed breakpoints with GGPL.

#### C.1.2 OTHER BASELINE MODELS FOR COMPARISON

To establish a comprehensive performance benchmark, we compare our GGPL-enhanced models against the following groups of baseline models.

- **GBDT models:** These models represent the traditional machine learning methods for tabular data.
  XGBoost (Chen & Guestrin, 2016), LightGBM (Ke et al., 2017), CatBoost (Prokhorenkova et al., 2018)

- **Other Deep Learning Models:** These models represent diverse advancements in deep learning architectures for tabular data.

Table 10: Detailed characteristics of the 46 benchmark datasets.

| Dataset | # Samples | # Feat. | # Num | # Bin | # Cat | Task | # Classes | Reference |
|---|---|---|---|---|---|---|---|---|
| Adult | 48842 | 14 | 6 | 1 | 7 | cls. | 2 | Gorishniy et al. (2024) |
| Black_Friday | 166821 | 9 | 4 | 1 | 4 | reg. | - | Gorishniy et al. (2024) |
| California_Housing | 20640 | 8 | 8 | 0 | 0 | reg. | - | Gorishniy et al. (2024) |
| Churn_Modelling | 10000 | 11 | 7 | 3 | 1 | cls. | 2 | Gorishniy et al. (2024) |
| Covertype | 581012 | 15 | 10 | 4 | 1 | cls. | 7 | Gorishniy et al. (2024) |
| Diamond | 53940 | 9 | 6 | 0 | 3 | reg. | - | Gorishniy et al. (2024) |
| Higgs_Small | 98049 | 28 | 28 | 0 | 0 | cls. | 2 | Gorishniy et al. (2024) |
| House_16H | 22784 | 16 | 16 | 0 | 0 | reg. | - | Gorishniy et al. (2024) |
| Microsoft | 1200192 | 136 | 131 | 5 | 0 | reg. | - | Gorishniy et al. (2024) |
| Otto_Group_Products | 61878 | 93 | 93 | 0 | 0 | cls. | 9 | Gorishniy et al. (2024) |
| Ailerons | 13750 | 33 | 33 | 0 | 0 | reg. | - | Grinsztajn et al. (2022) |
| analcatdata_supreme | 4052 | 7 | 2 | 5 | 0 | reg. | - | Grinsztajn et al. (2022) |
| bank-marketing | 10578 | 7 | 7 | 0 | 0 | cls. | 2 | Grinsztajn et al. (2022) |
| Brazilian_houses | 10692 | 11 | 8 | 2 | 1 | reg. | - | Grinsztajn et al. (2022) |
| cpu_act | 8192 | 21 | 21 | 0 | 0 | reg. | - | Grinsztajn et al. (2022) |
| credit | 16714 | 10 | 10 | 0 | 0 | cls. | 2 | Grinsztajn et al. (2022) |
| elevators | 16599 | 16 | 16 | 0 | 0 | reg. | - | Grinsztajn et al. (2022) |
| fifa | 18063 | 5 | 5 | 0 | 0 | reg. | - | Grinsztajn et al. (2022) |
| house_sales | 21613 | 17 | 15 | 2 | 0 | reg. | - | Grinsztajn et al. (2022) |
| isolet | 7797 | 613 | 613 | 0 | 0 | reg. | - | Grinsztajn et al. (2022) |
| jannis | 57580 | 54 | 54 | 0 | 0 | cls. | 2 | Grinsztajn et al. (2022) |
| kdd_ipums_la_97-small | 5188 | 20 | 20 | 0 | 0 | cls. | 2 | Grinsztajn et al. (2022) |
| KDDCup09_upselling | 5032 | 49 | 34 | 1 | 14 | cls. | 2 | Grinsztajn et al. (2022) |
| MagicTelescope | 13376 | 10 | 10 | 0 | 0 | cls. | 2 | Grinsztajn et al. (2022) |
| medical_charges | 163065 | 3 | 3 | 0 | 0 | reg. | - | Grinsztajn et al. (2022) |
| Mercedes_Benz | 4209 | 359 | 0 | 356 | 3 | reg. | - | Grinsztajn et al. (2022) |
| MiamiHousing2016 | 13932 | 13 | 13 | 0 | 0 | reg. | - | Grinsztajn et al. (2022) |
| MiniBooNE | 72998 | 50 | 50 | 0 | 0 | cls. | 2 | Grinsztajn et al. (2022) |
| nyc-taxi-green | 581835 | 16 | 9 | 3 | 4 | reg. | - | Grinsztajn et al. (2022) |
| OnlineNewsPopularity | 39644 | 59 | 45 | 14 | 0 | reg. | - | Grinsztajn et al. (2022) |
| particulate-matter-ukair | 394299 | 6 | 3 | 0 | 3 | reg. | - | Grinsztajn et al. (2022) |
| phoneme | 3172 | 5 | 5 | 0 | 0 | cls. | 2 | Grinsztajn et al. (2022) |
| pol | 15000 | 26 | 26 | 0 | 0 | reg. | - | Grinsztajn et al. (2022) |
| road-safety | 111762 | 32 | 29 | 0 | 3 | cls. | 2 | Grinsztajn et al. (2022) |
| superconduct | 21263 | 79 | 79 | 0 | 0 | reg. | - | Grinsztajn et al. (2022) |
| wine | 2554 | 11 | 11 | 0 | 0 | cls. | 2 | Grinsztajn et al. (2022) |
| wine_quality | 6497 | 11 | 11 | 0 | 0 | reg. | - | Grinsztajn et al. (2022) |
| year | 515345 | 90 | 90 | 0 | 0 | reg. | - | Grinsztajn et al. (2022) |
| Cooking_Time | 319986 | 192 | 186 | 3 | 3 | reg. | - | Rubachev et al. (2025) |
| Delivery_ETA | 350516 | 220 | 218 | 1 | 1 | reg. | - | Rubachev et al. (2025) |
| Ecom_Offers | 160057 | 110 | 104 | 6 | 0 | cls. | 2 | Rubachev et al. (2025) |
| Homecredit_Default | 381664 | 677 | 593 | 2 | 82 | cls. | 2 | Rubachev et al. (2025) |
| Homesite_Insurance | 260753 | 298 | 252 | 23 | 23 | cls. | 2 | Rubachev et al. (2025) |
| Maps_Routing | 279945 | 986 | 984 | 0 | 2 | reg. | - | Rubachev et al. (2025) |
| Sberbank_Housing | 28321 | 392 | 365 | 17 | 10 | reg. | - | Rubachev et al. (2025) |
| Weather | 189963 | 99 | 96 | 3 | 0 | reg. | - | Rubachev et al. (2025) |

SNN (Klambauer et al., 2017), FT-Transformer (Gorishniy et al., 2021), SAINT (Somepalli et al., 2021), DCN2 (Wang et al., 2021), Trompt (Chen et al., 2023), ExcelFormer (Chen et al., 2024), TabR (Gorishniy et al., 2024)

- **Foundation Model:** TabPFN is a pre-trained foundation model that can perform inference on unseen tasks without any parameter tuning, although its application is limited by dataset size constraints.

  TabPFN (Hollmann et al., 2025)

## C.2 IMPLEMENTATION DETAILS

### C.2.1 HARDWARE ENVIRONMENT

Our experiments were conducted on servers equipped with Intel(R) Xeon(R) Gold 6240 CPUs @ 2.60GHz and NVIDIA RTX 3090 GPUs. While most experiments were run on a single GPU, training on large datasets required up to 8 GPUs to meet GPU memory demands, particularly for models with high memory consumption (T2G-Former Yan et al., 2023, ModernNCA Ye et al., 2025).

### C.2.2 HYPERPARAMETER SEARCH SPACES

All models are trained using the AdamW optimizer (Loshchilov & Hutter, 2019) with an early stopping patience of 16 epochs, and hyperparameters were tuned using Optuna (Akiba et al., 2019) with the TPE sampler over 100 trials for most datasets, and 50 trials for large ones.

For our GGPL-enhanced backbone models (MLP, T2G-Former, ModernNCA, and TabM), the search spaces for all non-GGPL hyperparameters were kept identical to those in Liu et al. (2024) for T2G-Former and ModernNCA and those in Gorishniy et al. (2025) for MLP and TabM. We provide the detailed search spaces in Tables 11 to 14.

For all baseline models, including GBDTs and other deep learning methods, the hyperparameter search spaces were kept identical to those defined in Gorishniy et al. (2025). We refer to their original paper for the details.

## D   DETAILED EXPERIMENTAL RESULTS

Tables 15 and 16 provide the detailed performance metrics for all models across all 46 datasets, including the mean and standard deviation over 15 random seeds. For the baseline models, we report the performance scores directly from the original benchmark publication by Gorishniy et al. (2025), as our experimental setup is identical to theirs.

Table 11: Hyperparameter search spaces for MLP-GGPL.

| Hyperparameter | Search Space |
|---|---|
| Learning rate | LogUniform: [3e-5, 0.001] |
| Weight decay | {0, LogUniform: [0.0001, 0.1]} |
| # layers | Int: [1, 5] |
| Width | Int: [64, 1024, 16] |
| Dropout | {0, Uniform: [0.0, 0.5]} |
| Embedding dim. (d) | Int: [8, 32, 4] |
| Average number of breakpoints (K) | Int: [2, 48] |
| Deactivation Prob. (p) | {0, Uniform: [0.0, 0.3]} |

Table 12: Hyperparameter search spaces for T2G-Former-GGPL.

| Hyperparameter | Search Space |
|---|---|
| Learning rate | LogUniform: [1e-5, 0.001] |
| Weight decay | LogUniform: [1e-6, 0.001] |
| # layers | Int: [1, 4] |
| Token width | Categorical: {8, 16, 32, 64, 128} |
| Residual dropout | {0, Uniform: [0.0, 0.2]} |
| Attention dropout | Uniform: [0.0, 0.5] |
| FFN dropout | Uniform: [0.0, 0.5] |
| FFN expansion rate | Uniform: [0.67, 2.67] |
| Frozen switch | Categorical: {true, false} |
| Activation | reglu |
| Num heads | 8 |
| Embedding dim. (d) | Identical to the token width |
| Average number of breakpoints (K) | Int: [2, 48] |
| Deactivation Prob. (p) | {0, Uniform: [0.0, 0.3]} |

Table 13: Hyperparameter search spaces for ModernNCA-GGPL.

| Hyperparameter | Search Space |
| --- | --- |
| Learning rate | LogUniform: [1e-5, 0.1] |
| Weight decay | {0, LogUniform: [1e-6, 0.001]} |
| # MLP layers | {0, Int: [0, 2]} |
| MLP width | Int: [64, 1024] |
| Projection Dim. | Int: [64, 1024] |
| Dropout | Uniform: [0.0, 0.5] |
| Sample rate | Uniform: [0.05, 0.6] |
| Temperature | 1.0 |
| Embedding dim. (d) | Int: [8, 32, 4] |
| Average number of breakpoints (K) | Int: [2, 48] |
| Deactivation Prob. (p) | {0, Uniform: [0.0, 0.3]} |

Table 14: Hyperparameter search spaces for TabM-mini-GGPL.

| Hyperparameter | Search Space |
| --- | --- |
| Learning rate | LogUniform: [0.0001, 0.003] |
| Weight decay | {0, LogUniform: [0.0001, 0.1]} |
| # layers | Int: [1, 4] |
| Width | Int: [64, 1024, 16] |
| Dropout | {0, Uniform: [0.0, 0.5]} |
| # ensembles | 32 |
| Embedding dim. (d) | Int: [8, 32, 4] |
| Average number of breakpoints (K) | Int: [2, 48] |
| Deactivation Prob. (p) | {0, Uniform: [0.0, 0.3]} |

Table 15: Detailed classification results (Accuracy ↑). Scores for baseline models are from Gorishniy et al. (2025).

| Dataset | LightGBM | XGBoost | CatBoost | DCN2 | SNN | Excel | SAINT | FT-T | Trompt | MLP | T2G | TabR | MNCA | TabM | TabPFN | MLP-GGPL | T2G-GGPL | MNCA-GGPL | TabM-GGPL |
|---|---|---|---|---|---|---|---|---|---|---|---|---|---|---|---|---|---|---|---|
| Adult | | | | | | | | | | | | | | | | | | | |
| Churn_Modelling | | | | | | | | | | | | | | | | | | | |
| Covertype | | | | | | | | | | | | | | | | | | | |
| Higgs_Small | | | | | | | | | | | | | | | | | | | |
| Otto_Group_Products | | | | | | | | | | | | | | | | | | | |
| Ecom_Offers | | | | | | | | | | | | | | | | | | | |
| Homecredit_Default | | | | | | | | | | | | | | | | | | | |
| Homesite_Insurance | | | | | | | | | | | | | | | | | | | |
| bank-marketing | | | | | | | | | | | | | | | | | | | |
| credit | | | | | | | | | | | | | | | | | | | |
| janus | | | | | | | | | | | | | | | | | | | |
| kdd_ipums_la_97-small | | | | | | | | | | | | | | | | | | | |
| KDDCup09_upselling | | | | | | | | | | | | | | | | | | | |
| MagicTelescope | | | | | | | | | | | | | | | | | | | |
| MiniBooNE | | | | | | | | | | | | | | | | | | | |
| phoneme | | | | | | | | | | | | | | | | | | | |
| road-safety | | | | | | | | | | | | | | | | | | | |
| wine | | | | | | | | | | | | | | | | | | | |

Table 16: Detailed regression results (RMSE ↓). Scores for baseline models are from Gorishniy et al. (2025).

| Dataset | LightGBM | XGBoost | CatBoost | DCN2 | SNN | Excel | SAINT | FT-T | Trompt | MLP | T2G | TabR | MNCA | TabM | TabPFN | MLP-GGPL | T2G-GGPL | MNCA-GGPL | TabM-GGPL |
|---|---|---|---|---|---|---|---|---|---|---|---|---|---|---|---|---|---|---|---|
| Black_Friday | | | | | | | | | | | | | | | | | | | |
| California_Housing | | | | | | | | | | | | | | | | | | | |
| Diamond | | | | | | | | | | | | | | | | | | | |
| House_16H (×1e-4) | | | | | | | | | | | | | | | | | | | |
| Microsoft | | | | | | | | | | | | | | | | | | | |
| Cooking_Time | | | | | | | | | | | | | | | | | | | |
| Delivery_ETA | | | | | | | | | | | | | | | | | | | |
| Maps_Routing | | | | | | | | | | | | | | | | | | | |
| Sberbank_Housing | | | | | | | | | | | | | | | | | | | |
| Weather | | | | | | | | | | | | | | | | | | | |
| Ailerons (×1e3) | | | | | | | | | | | | | | | | | | | |
| analcatdata_supreme | | | | | | | | | | | | | | | | | | | |
| Brazilian_houses | | | | | | | | | | | | | | | | | | | |
| cpu_act | | | | | | | | | | | | | | | | | | | |
| elevators | | | | | | | | | | | | | | | | | | | |
| fifa | | | | | | | | | | | | | | | | | | | |
| isolet | | | | | | | | | | | | | | | | | | | |
| house_sales | | | | | | | | | | | | | | | | | | | |
| medical_charges | | | | | | | | | | | | | | | | | | | |
| Mercedes_Benz_Greener_Manufacturing | | | | | | | | | | | | | | | | | | | |
| MiamiHousing2016 | | | | | | | | | | | | | | | | | | | |
| nyc-taxi-green-dec-2016 | | | | | | | | | | | | | | | | | | | |
| OnlineNewsPopularity | | | | | | | | | | | | | | | | | | | |
| particulate-matter-ukair-2017 | | | | | | | | | | | | | | | | | | | |
| pol | | | | | | | | | | | | | | | | | | | |
| superconduct | | | | | | | | | | | | | | | | | | | |
| wine_quality | | | | | | | | | | | | | | | | | | | |
| year | | | | | | | | | | | | | | | | | | | |

# E    DETAILED ANALYSIS

## E.1    ELO EVALUATION DETAILS

We use an Elo-based evaluation Elo (1967) to summarize the relative performance of multiple archi-
tectures across heterogeneous datasets and tasks that differ in both evaluation metrics and difficulty.
We include all $18$ models considered in our main experiments (Table 2). These consist of each
backbone (MLP, T2G-Former, ModernNCA, TabM-mini) with its native numerical embedding and
with the proposed GGPL embedding, together with additional baselines such as GBDTs. The Elo
evaluation is computed jointly over all $46$ datasets (classification and regression), using the same
metrics as in the main experiments (accuracy for classification and RMSE for regression).

Following TabArena (Erickson et al., 2025), we use a stable Bradley-Terry implementation to com-
pute Elo scores and 200 bootstrapping rounds to approximate $2.5\%$-$97.5\%$ confidence interval. We
also adopt the 400-point Elo gap. The expected win rate of $i$-th model with Elo ratings of $R_i$ against
$R_j$ is

$$E_i = \frac{1}{1 + 10^{(R_j - R_i)/400}},$$

Finally, unlike TabArena, we calibrate 1000 Elo to the performance of native MLP model.

Figure 5 summarizes the resulting Elo ratings and confidence intervals for all 18 models. GGPL-
enhanced variants consistently achieve higher Elo scores than their native counterparts, confirming
the improvements reported in the main text.

## E.2    DEFINITION OF EMBEDDING TORTUOSITY

We quantify the complexity of a learned numerical embedding ($\phi$) by measuring its tortuosity. For
a given numerical feature $i$, let $t_1^{(i)} < t_2^{(i)} < \cdots < t_{K_i}^{(i)}$ denote the breakpoints that are initial-
ized from GBDT (e.g., XGBoost) splits and subsequently optimized by backpropagation, and let
$\mathbf{v}_1^{(i)}, \ldots, \mathbf{v}_{K_i}^{(i)} \in \mathbb{R}^d$ be the corresponding embedding vectors. As formulated in Eq. 4, we define the
tortuosity of $\phi_i$ as the ratio of the piecewise-linear curve length to the Euclidean distance between
its start and end points. A higher value indicates a more complicated function, while a value closer
to 1 signifies a smoother path.

$$\text{Tortuosity}(\phi_i) = \frac{\sum_{k=0}^{K_i} \left\| [\mathbf{v}_{k+1}^{(i)}; t_{k+1}^{(i)}] - [\mathbf{v}_k^{(i)}; t_k^{(i)}] \right\|_2}{\left\| [\mathbf{v}_{K_i+1}^{(i)}; t_{K_i+1}^{(i)}] - [\mathbf{v}_0^{(i)}; t_0^{(i)}] \right\|_2} \tag{4}$$

## E.3    EFFECT SIZE DETAILS

We aggregate effect sizes across datasets by normalizing each dataset so that the best-performing
model attains a score of 1 and the worst-performing model a score of 0. For each backbone and task,
we then define the normalized-score gain $\Delta$ as the difference between the GGPL-enhanced variant
and its baseline (native) numerical embedding, and summarize this gain via its mean and standard
deviation over all datasets and seeds.

Figure 6 shows histograms of these normalized scores for regression (left) and classification (right)
across all datasets and seeds for the four backbones (TabM-mini, ModernNCA, T2G-Former, MLP).
For each architecture, we compare the native numerical embedding (Baseline) and GGPL, along
with the mean $\mu(\Delta)$ and standard deviation $\sigma(\Delta)$ of $\Delta$. On regression tasks, all backbones exhibit
positive mean gains, and the GGPL histograms are shifted toward 1 relative to their baselines, indi-
cating consistent improvements in performance. On classification tasks, the two histograms overlap.
Table 2 and Figure 6 confirm that GGPL consistently improves regression performance and yields
slightly improved, at least comparable performance on classification.

Figure 5: Elo ratings with 95% confidence intervals for all 18 models considered in Table 2.

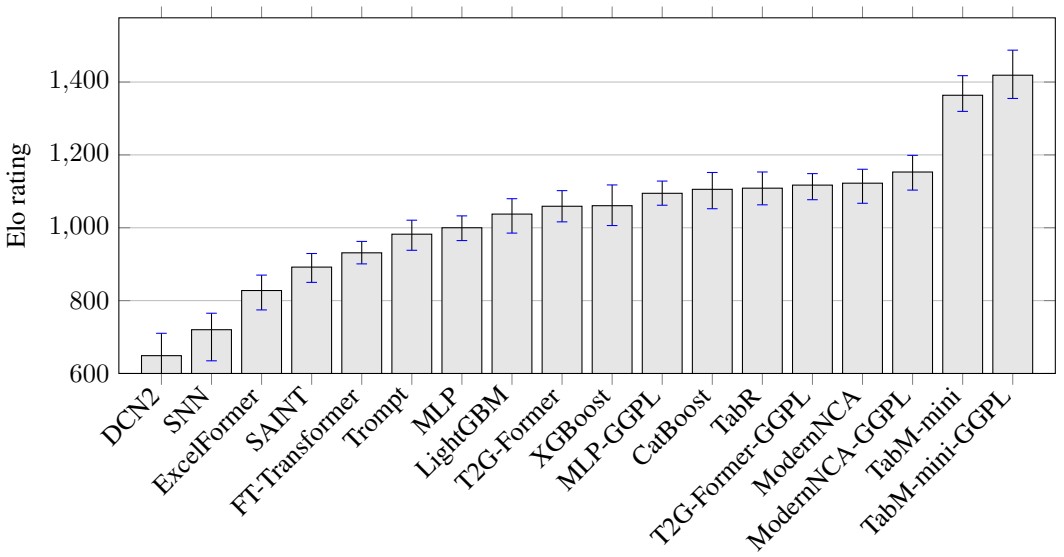

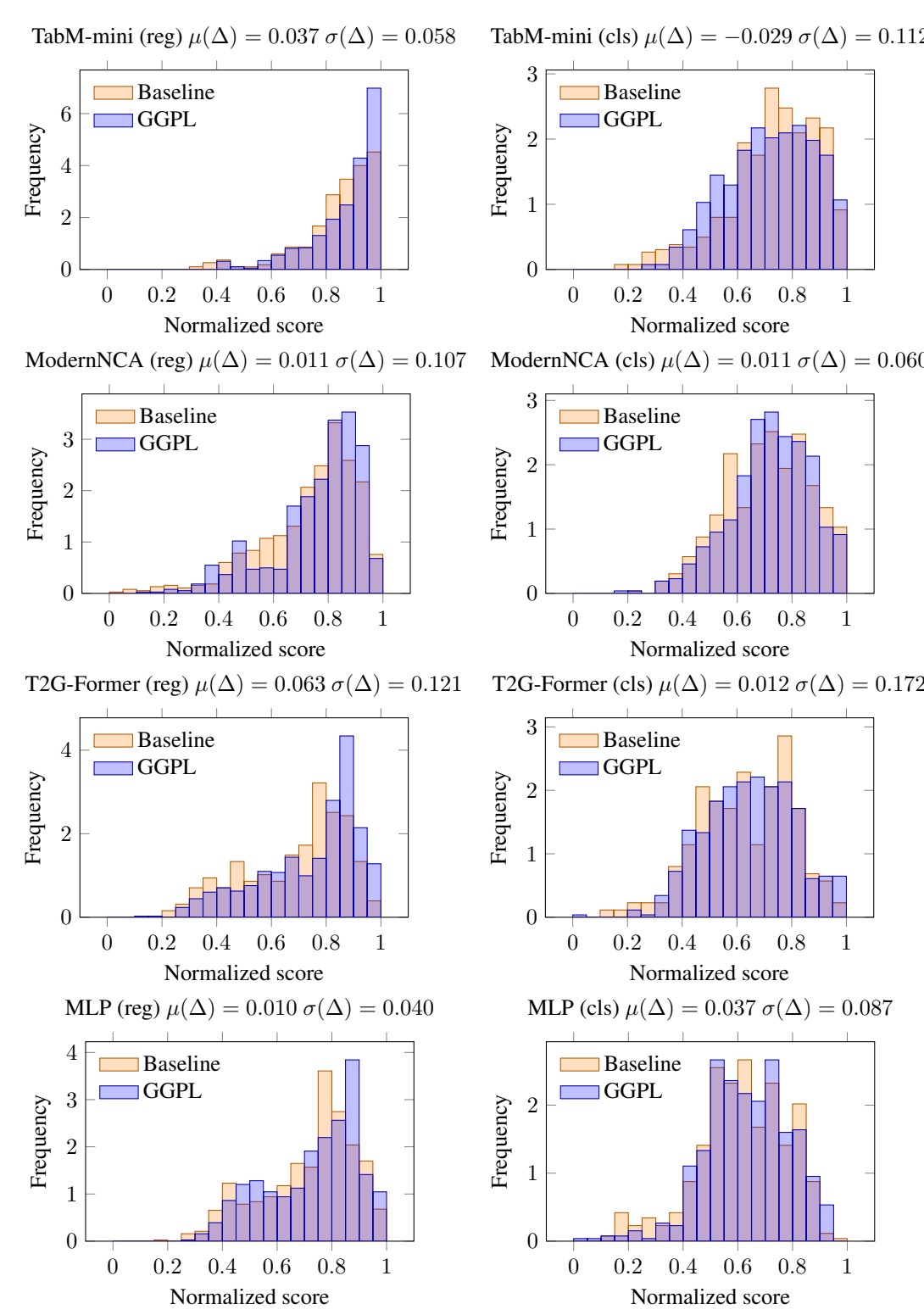

Figure 6: Histograms of normalized scores for regression (left) and classification (right) across all datasets and seeds for four backbones (TabM-mini, ModernNCA, T2G-Former, MLP). For each architecture we compare the native numerical embedding (Baseline) and GGPL, and report the mean $\mu(\Delta)$ and standard deviation $\sigma(\Delta)$ of the normalized-score gain $\Delta$.

