# OpenReview forum: "Enhancing Deep Tabular Models with GBDT-Guided Piecewise-Linear Embeddings"
_ICLR.cc/2026/Conference — Submitted to ICLR 2026_

### Official Review · Reviewer_AdEJ · 2025-10-17

**Soundness:** 3
**Presentation:** 3
**Contribution:** 2
**Rating:** 2
**Confidence:** 4

**Summary:**

The paper proposes GBDT-Guided Piecewise-Linear (GGPL) embeddings for numerical features in tabular DL. The recipe: (i) initialize breakpoints from XGBoost split thresholds; (ii) optimize breakpoint locations via a simplex reparameterization; (iii) apply stochastic breakpoint deactivation as a regularizer (used only for regression). On 46 datasets, the authors report average-rank gains when swapping GGPL into several backbones (MLP, T2G-Former, ModernNCA, TabM-mini), with TabM-mini-GGPL achieving the best overall average rank (2.96) and small but consistent ablation gains over a uniform PLE baseline. They also present Wilcoxon tests against alternative numerical embeddings and against native embeddings in other backbones, and a sensitivity study showing default XGBoost is “not worse” than tuned (p=0.067)

**Strengths:**

Tidy, drop-in module; clear problem framing around numerical embeddings. The method is simple and training-only, with no stated inference-time overhead relative to prior PLEs.

Broad evaluation (46 datasets, 15 seeds), with dataset characteristics analysis and ablations (I/O/R).

Statistical testing across seeds/datasets is a plus (stratified Wilcoxon).

**Weaknesses:**

1. Using tree thresholds to seed piecewise segments is very close in spirit to target-aware or supervised breakpoint selection used in prior PLE work; the key twist is to harvest splits from a GBDT. The simplex reparameterization is a standard trick to enforce ordered breakpoints, and “stochastic breakpoint deactivation” is conceptually akin to dropout over segments. The paper reads as engineering polish on known ingredients rather than a conceptually new embedding family.

2. Main claims are summarized via average ranks over mixed tasks and datasets (Table 2). Ranks can hide small absolute gains and do not convey practical effect sizes. Some gains are modest (e.g., MLP: rank 10.32→7.64 for regression; classification gains are ~1 rank or worse for some backbones), and T2G-Former’s classification rank slightly degrades with GGPL (8.83→8.94). Absolute accuracy/RMSE deltas (with CIs) should be aggregated and reported.

3. GGPL introduces extra knobs (total breakpoints K, deactivation p) with wide search spaces (up to 48× the number of numeric features), which effectively increases capacity and tuning surface relative to some baselines’ numerical encoders. While authors state non-GGPL hyperparameter spaces were kept identical to prior work, GGPL-specific spaces add degrees of freedom; it’s unclear whether baseline numerical embeddings received equivalently rich capacity/tuning to match. A capacity-matched control (e.g., target-aware PLE with the same K sweep) would be important.

4. Despite “no inference overhead,” the training pipeline requires fitting an auxiliary XGBoost and tuning extra GGPL hyperparameters (and the overall protocol uses Optuna with 100 trials on most datasets). The paper does not quantify training-time/energy overheads vs. baselines. The “default XGBoost is fine” claim hinges on p=0.067 (borderline), which does not strongly support robustness of the initializer.

5. The stochastic breakpoint regularizer is deliberately disabled for classification, and the improvements there are minor or negative in places (e.g., T2G-Former). This undercuts the “effective across tasks” narrative.

6. Beyond periodic/PLE/T2V, the paper omits comparisons to other learnable, high-frequency-friendly encoders (e.g., spline-based or lattice-based numerical transforms). Even one strong representative would help calibrate whether GGPL’s benefits stem from smarter breakpoints or just more flexible piece counts. (The paper’s own framing highlights high-frequency target components as the motivation.)

**Questions:**

Please report aggregate effect sizes (median % improvement and mean normalized deltas) alongside ranks; include 95% CIs across seeds/datasets. Can you share per-task (regression vs classification) absolute improvements averaged across datasets? (You already have Tables 14–15—aggregate them.)

Provide training time/compute comparisons (including the XGBoost prepass and Optuna trials) vs. target-aware PLE and periodic embeddings under matched budgets.

Run a capacity-matched control: target-aware PLE with the same K search as GGPL across backbones, and report whether GGPL still wins.

Add at least one spline/lattice numerical encoder baseline to test the “high-frequency” rationale.

For classification, try a breakpoint regularizer that doesn’t enforce smoothness (e.g., piece-drop with re-weighting rather than linear interpolation) or show that disabling R is indeed optimal across datasets.

---

> ### Author Response · Authors · 2025-11-21
>
> Thank you for the detailed and constructive feedback, and for pointing out several places where we could clarify the design and evaluation of GGPL.
>
> **Hyperparameters, capacity, and tuning protocol (Weaknesses 3, 4, 6; Questions 2, 3, 4).**
>
> We apologize for the ambiguity in the hyperparameter description and have revised the text accordingly (App. C.2). The number of hyperparameters used by GGPL is similar to baselines: quantile-based PLE uses 2, target-aware PLE uses 4, and periodic embeddings use 3; GGPL uses 2 for classification and 3 for regression. We use one hyperparameter to determine the average number of breakpoints of all numerical features once, rather than tuning them for each features individually.
>
> We also clarify that the search space and capacity are smaller than in prior PLEs. Previous PLEs search $d_{\text{emb}} \in \text{Int}[8, 32]$ and $n_{\text{bins}} \in \text{Int}[2, 128]$ and allocate the same number of bins to every numerical feature. In contrast, GGPL uses the same $d_{\text{emb}} \in \text{Int}[8, 32]$ for all numerical features and searches $n_{\text{bins}} \in \text{Int}[2, 48]$ for the average number of segments, which are then distributed non-uniformly across features.
>
> We also clarify that, for fairness, all baselines are tuned with the same HPO trial budget as GGPL. Also, fitting a default XGBoost model is performed only once per dataset and takes about 1 second on a moderate-sized dataset (50,000 samples). This cost is negligible compared to tuning the MLP, where training takes about 1 minute per HPO trial and we typically run several dozen trials per dataset.
>
> **Positioning relative to prior PLE work (Weakness 1).**
>
> We agree that our method is closely related to previous piecewise-linear embeddings. Our contribution is to revisit this framework and highlight the importance of breakpoint locations, which we argue has been relatively underexplored since Gorishniy et al. (2022). We now emphasize more clearly that our contribution is to provide an efficient and practical way to place breakpoints.
>
> **Default vs tuned XGBoost initializer (part of Weakness 4).**
>
> We agree that tuned XGBoost tends to provide slightly better breakpoints than default XGBoost, and we do not view this as a weakness. Our goal is to provide a practical default numerical embedding with minimal additional training pipeline overhead, as you pointed out in strength 1. To this end, we use default XGBoost as the initializer in our main experiments. As we show in Sec. 3.3 and App. A.2, tuned XGBoost can yield modest additional gains over default XGBoost (with $p \approx 0.067$), which practitioners may exploit if they are willing to pay the extra tuning cost and pipeline overhead.
>
> **Magnitude and consistency of improvements (Weakness 2, Question 1).**
>
> We added two analyses to better quantify the improvement beyond average ranks.
>
> (i) **Elo analysis** (Sec. 5.2, App. E.1): following TabArena [1,2], we compute Elo ratings with 95% CIs for all 18 models. In all four backbones, the GGPL variants achieve higher Elo than their respective baselines.
>
> (ii) **Normalized-score gains** (Sec. 5.5, App. E.3): to aggregate datasets of different difficulties, instead of raw improvement we normalize each dataset so that the best model scores 1 and the worst scores 0. As in our per-dataset tables, gains are consistently positive on regression, while on classification GGPL is overall similar or slightly better than the native embeddings.
>
> Enhancing numerical embeddings is a key objective because numerical features constitute the primary information channel in tabular data, yet they remain largely ad hoc in current architectures. A more expressive but lightweight numerical embedding can therefore yield broad, architecture-agnostic improvements. Our results indicate that GGPL provides such a practical default with minimal overhead.
>
> **Effect of stochastic breakpoint regularization (Weakness 5, Question 5).**
>
> We expanded our analysis of stochastic breakpoint regularization in Sec. 3.5 and App. A.3. The stratified Wilcoxon test shows a positive trend in favor of the regularized variant on regression tasks ($Z > 0$). On classification, however, the test statistic is slightly negative with a large $p$-value ($p \approx 0.87$), indicating that this particular regularizer does not help for classification. Based on these results we apply stochastic breakpoint regularization only to regression tasks. Exploring alternative regularizers that do not enforce smoothness (e.g., piece-drop variants) for classification is an interesting direction that we leave for future work.

---

> > ### Author Response · Authors · 2025-11-21
> >
> > **Comparison to spline/lattice encoders (Weakness 6, Question 4).**
> >
> > We appreciate the suggestion to consider spline- or lattice-based numerical encoders. However, from a functional perspective, piecewise-linear embeddings is a first-order splines, and increasing the spline order mainly smooths the function rather than sharpening it. In addition, as clarified above, the search range for the number of segments is actually narrower than prior PLEs (we use $n_{\text{bins}} \in \text{Int}[2, 48]$ vs. $\text{Int}[2, 128]$ in previous PLEs). In this setting, the observed improvements reflect the effect of breakpoint placement rather than increased piece counts. Nonetheless, a broader comparison to spline- and lattice-based transforms remains an interesting direction for future work.
> >
> > [1] Erickson, Nick, et al. "TabArena: A Living Benchmark for Machine Learning on Tabular Data." arXiv preprint arXiv:2506.16791 (2025).
> >
> > [2] Grinsztajn, Léo, et al. "TabPFN-2.5: Advancing the State of the Art in Tabular Foundation Models." *arXiv preprint arXiv:2511.08667* (2025).

---

### Official Review · Reviewer_pVDs · 2025-10-31

**Soundness:** 3
**Presentation:** 3
**Contribution:** 3
**Rating:** 8
**Confidence:** 2

**Summary:**

The authors presented a novel deep learning method for tabular data prediction
The method revolves around embedding numerical features so that they are easier for the deep learning method to use
Building upon previous methods of using fixed breakpoints of piece wise linear embeddings, the authors used gradient boosted decision trees to determine the breakpoints.
The authors showed that the proposed method had better performance compared to not using the GGPL embedding in numerous models.

**Strengths:**

* Paper is clear and had direct extension of previous work
* GGPL embedding is versatile and could easily be applied to any deep learning architecture
* Results showed clear improvements in performance improvements across multiple models regardless of their architecture

**Weaknesses:**

* Discretizing numeric features into piecewise linear has been explored in previous works [Gorishniy 2022]. The only innovation is to define the breakpoints with gradient boosted trees
* Since the discretization is dependent on gradient boosted trees, it might be interesting to understand the performance and memory requirements changes in different datasets. In particular datasets with numerical features with a wide range

**Questions:**

* Do different gradient boosted tree methods alter performance?
* The final snapshot of the trained XGBoost method was used to get the breakpoints, were there explorations to check the intermediate trees to see if there any possible performance boosts?

---

> ### Author Response · Authors · 2025-11-21
>
> Thank you for the constructive suggestions on clarifying our contribution and analysis.
>
> **Novelty of breakpoint selection (Weakness 1).**
>
> We agree that our method is built upon Gorishniy et al. (2022). Our contribution is to revisit this framework and demonstrate that the choice of breakpoint locations—an aspect that has received limited attention since the original work—is central to the effectiveness of piecewise-linear numerical embeddings.
>
> **Memory requirements and dependence on feature range (Weakness 2).**
>
> The total number of segments is chosen from a hyperparameter range similar to prior piecewise-linear embeddings, so we do not expect substantial changes in memory requirements. Additionally, since numerical features are normalized before embedding, wide raw ranges have limited direct impact. There may be cases where highly important features receive more breakpoints than others, leading to an unbalanced allocation across dimensions. However, compared to the baseline that allocates an equal number of breakpoints to each feature, our non-uniform allocation is expected to be more efficient in those cases.
>
> **Effect of GBDT initializer choice (Questions 1, 2).**
>
> We added an analysis of different GBDT initializations in Sec. 3.3 and App. A.2. Overall, stronger GBDT models tend to give slightly better downstream performance, but the effect is not statistically significant at standard levels (e.g., tuned XGBoost vs. default XGBoost yields $p \approx 0.07$). Default XGBoost and default LightGBM perform similarly. Finally, we varied the number of trees in XGBoost to approximate intermediate snapshots; as long as the number of trees is not too small, we observed minor differences in downstream performance.

---

### Official Review · Reviewer_LZRQ · 2025-11-01

**Soundness:** 2
**Presentation:** 3
**Contribution:** 2
**Rating:** 4
**Confidence:** 3

**Summary:**

This paper tackles the challenge of learning expressive and robust representations for numerical features in deep tabular models. While categorical features are effectively handled through embedding layers, numerical embeddings remain underexplored. The authors introduce an elegant and effective approach, GBDT-Guided Piecewise-Linear (GGPL) embeddings, which integrates three components: initialization of breakpoints using GBDT (XGBoost) split thresholds, stable optimization via simplex-based reparameterization to preserve order, and stochastic breakpoint regularization to prevent overfitting. Comprehensive experiments on multiple tabular datasets demonstrate that GGPL consistently enhances the performance of state-of-the-art tabular models without adding inference-time cost.

**Strengths:**

1. The paper tackles an important and timely problem in deep learning for tabular data, where numerical feature embeddings remain underexplored.
2. The writing is clear and well-organized, making the paper easy to follow and accessible to both deep learning and tabular data researchers.
3. The experimental setup is comprehensive, covering multiple datasets and multiple baselines, which demonstrates the generality and robustness of the proposed approach.

**Weaknesses:**

1. The paper appears to make several overclaims regarding novelty. In particular, Contribution 1 (lines 80–82) states that the proposed method introduces piecewise-linear embeddings for numerical features, but this idea has already been explored in prior work, notably by Yury Gorishniy et al., “On Embeddings for Numerical Features in Tabular Deep Learning,” NeurIPS 2022. While the authors cite this paper, the Introduction section (e.g., lines 88–90) does not sufficiently acknowledge or discuss the methodological overlap and differences. I recommend explicitly positioning this work relative to Gorishniy et al. (2022), clarifying what is truly novel (e.g., the GBDT-guided initialization or stochastic regularization), to avoid the impression of rediscovery.
2. Could you clarify how XGBoost and CatBoost were implemented? Were they run with default hyperparameters  (as in lines 224–230) or tuned per dataset? If tuned, could you report detailed parameters per dataset?

**Questions:**

I have questions about the implementation of tree-based method. see above

---

> ### Author Response · Authors · 2025-11-21
>
> Thank you for the helpful feedback and for pointing out where our novelty and experimental details could be clarified.
>
> **Novelty and relation to prior work (Weakness 1).**
>
> We revised the Introduction, especially the contribution paragraph, to more clearly position our work relative to Gorishniy et al. (2022). While their work introduced piecewise-linear numerical embeddings, our contribution is orthogonal: we highlight that the choice of breakpoint locations is a critical yet underexplored factor, and we propose an efficient procedure for setting them.  We also clarified that the goal of our paper is to provide a practical default numerical embedding for future tabular architectures with minimal overhead.
>
> **Implementation of XGBoost and CatBoost (Weakness 2, Question).**
>
> We clarified Sec. 4.3 and App. C.2 to specify that the XGBoost and CatBoost baselines are tuned per dataset, using the same hyperparameter search protocol as Gorishniy et al. (2025) for a fair comparison. We use default XGBoost only for breakpoint initialization within our method, in order to keep the additional overhead minimal.

---

### Official Review · Reviewer_wYfW · 2025-11-05

**Soundness:** 2
**Presentation:** 2
**Contribution:** 2
**Rating:** 4
**Confidence:** 3

**Summary:**

This paper proposes an improved numerical feature embedding scheme for tabular neural networks. Its design is based on a piecewise-linear embedding from [Gorishniy 2022](https://proceedings.neurips.cc/paper_files/paper/2022/hash/9e9f0ffc3d836836ca96cbf8fe14b105-Abstract-Conference.html). The authors propose to take the feature binning thresholds from maximal gain XGBoost tree splits instead of taking them uniformly or from a single-feature tree building process. The authors also propose a procedure to finetune the bin edges (based on a reparametrization that keeps the order intact). Authors also propose a dropout-like embedding regularization via dropping bin thresholds at random during training. The efficacy is evaluated on a benchmark from the [TabM 2025](https://arxiv.org/abs/2410.24210) paper and claims that the new "GGPL" embedding scheme maintains the best average rank when combined with SoTA tabular neural networks and is generally better than prior embedding variations.

**Strengths:**

The experiments are well done and trustworthy. Most of the relevant details for reproduction are present and the paper is clearly written. The motivation of creating a more universal (e.g. without the need to tune the frequency hyperparameter) embedding which is better than the PLE from prior work is also strong and I believe this is an important area for future tabular neural networks.

**Weaknesses:**

In my view the improvement and its consistency is not very compelling practically speaking. Looking at the raw results table in the appendix, the new embedding may produce slightly better results than the alternatives, but the magnitude of the improvement and its consistency is not good for the proposed complexity.

Another related weakness is the lack of explanations regarding where the effects come from (except some mentions that selecting bin edges is hard, prone to overfitting but helps with representational power). I suggest the authors look deeper into where the improvement actually comes from. I may suggest a few recent papers [1](https://arxiv.org/pdf/2509.04430), [2](https://arxiv.org/abs/2411.00247) that do this for tabular data specifically, maybe these perspectives could shed some more light onto why the proposed modifications help.

**Questions:**

- Can you provide average % improvement, as a means to judge the scale of the improvement brought by the new embedding scheme?
  - Looking through the results table in the appendix, it seems that most of the classification results are not significant when comparing TabM (I am assuming it uses the prior PLE variation) with TabM-GGPL. For regression problems improvement seems a bit more significant, but the relative gains are still marginal.
- Does dropout-like regularization really only help with regression? This claim seems a bit too high level for the usual state of affairs with tabular data (e.g. very very diverse set of datasets)
- How do you measure "Tortuosity" in Figure 3b?

---

> ### Author Response · Authors · 2025-11-21
>
> Thank you for the constructive feedback and for highlighting the importance of effect-size analysis and interpretability.
>
> **Magnitude and consistency of improvements (Weakness 1, Question 1).**
>
> We added two analyses to better quantify the improvement.
>
> (i) **Elo analysis** (Sec. 5.2, App. E.1): following TabArena [1,2], we compute Elo ratings with 95\% CIs for all 18 models. In all four backbones, the GGPL variants achieve higher Elo than their respective baselines.
>
> (ii) **Normalized-score gains** (Sec. 5.5, App. E.3): to aggregate datasets of different difficulties, instead of raw improvement we normalize each dataset so that the best model scores 1 and the worst scores 0. As you observed, gains are consistently positive on regression, while on classification GGPL is overall similar or slightly better than the native embeddings.
>
> Enhancing numerical embeddings is a key objective because numerical features constitute the primary information channel in tabular data, yet they remain largely ad hoc in current architectures. A more expressive but lightweight numerical embedding can therefore yield broad, architecture-agnostic improvements. Our results indicate that GGPL provides such a practical default with minimal overhead.
>
> **Where the improvement comes from (Weakness 2).**
>
> Thank you for the insightful suggestion. We agree that a full mechanistic account of why GGPL improves over prior numerical embeddings remains incomplete. This limitation reflects a broader gap in the literature, as the behavior of numerical embeddings in tabular models is still poorly understood. Our analysis in Sec. 5.3 provides initial empirical evidence isolating when GGPL is most beneficial, but developing a deeper theoretical explanation is an important and non-trivial direction for future work. We appreciate the references to recent interpretability studies and will incorporate these perspectives in follow-up research.
>
> **Effect of stochastic breakpoint regularization (Question 2).**
>
> We analyze the effect of regularization on different task types using all 46 datasets in Sec. 3.5 and App. A.3. On regression, the stratified Wilcoxon test shows a positive trend in favor of the regularized variant (Z > 0), whereas on classification the test statistic is slightly negative with a large $p$-value ($p \approx 0.87$), indicating that the regularization does not help for classification. We therefore apply stochastic breakpoint regularization only to regression tasks.
>
> **Definition of tortuosity (Question 3).**
>
> We added a definition in App. E.2 and refer to it in Sec. 5.4. Tortuosity is defined as the ratio between the length of the piecewise-linear curve obtained by following the embedding along adjacent breakpoints and the straight-line distance between the first and last embedding vectors.
>
> [1] Erickson, Nick, et al. "TabArena: A Living Benchmark for Machine Learning on Tabular Data." arXiv preprint arXiv:2506.16791 (2025).
>
> [2] Grinsztajn, Léo, et al. "TabPFN-2.5: Advancing the State of the Art in Tabular Foundation Models." *arXiv preprint arXiv:2511.08667* (2025).

---

### Meta-Review · Area_Chair_JdQS · 2025-12-19

**Summary:**

This submission proposes GGPL, a piecewise-linear embedding method for numerical features in deep tabular learning, integrating GBDT-guided breakpoint initialization, simplex-based optimization, and stochastic regularization.

Reviewers were concerned about the issues as follows:
- **Novelty**: GGPL’s key components are incremental adaptations of existing techniques rather than conceptually new contributions. The paper also inadequately distinguishes GGPL from prior piecewise-linear embedding (PLE) work (Gorishniy et al., 2022), risking the perception of rediscovery.
- **Marginal Effect**: Performance gains are limited and task-dependent. Classification tasks show negligible or even negative improvements (e.g., T2G-Former-GGPL rank degrades from 8.83 to 8.94), while regression gains, though more consistent, are modest in absolute terms.
- **Missing Analysis**: Critical experimental controls are missing. There is no capacity-matched comparison with baseline embeddings (e.g., target-aware PLE with the same breakpoint count search space), making it unclear if gains stem from GGPL’s design or increased model capacity.

**Reviewer Concerns:**

From the rebuttal, the authors have addressed part issue of **Marginal Effect** by adding Elo analysis with 95% CIs and normalized-score gains, along with Wilcoxon p-values. However, the authors failed to address the issues about **Novelty** and **Missing Analysis**.

**Reviewer Scores:**

Considering that there was not a sufficient discussion stage this year, it is difficult to see the reviewer's willingness to change the score from the review rebuttal stage of the discussion.

---

### Decision · Program_Chairs · 2026-01-26

Reject